# Measurement Invariance of a Direct Behavior Rating Multi Item Scale across Occasions

**Markus Gebhardt** [1,*], **Jeffrey M. DeVries** [1], **Jana Jungjohann** [1], **Gino Casale** [2], **Andreas Gegenfurtner** [3] **and Jörg-Tobias Kuhn** [4]

[1] Research in Inclusive Education, Faculty of Rehabilitation Science, Technical University of Dortmund, 44227 Dortmund, Germany; jeffrey.devries@tu-dortmund.de (J.M.D.); jana.jungjohann@tu-dortmund.de (J.J.)

[2] Department of Special Education, University of Cologne, 50931 Cologne, Germany; gino.casale@uni-koeln.de

[3] Deggendorf Institute of Technology, Institute for Quality and Continuing Education, 94469 Deggendorf, Germany; andreas.gegenfurtner@th-deg.de

[4] Faculty of Rehabilitation Science, Educational Research Methods, Technical University of Dortmund, 44227 Dortmund, Germany; tobias.kuhn@tu-dortmund.de

* Correspondence: markus.gebhardt@tu-dortmund.de; Tel.: +49-231-755-4546

**Abstract:** Direct Behavior Rating (DBR) as a behavioral progress monitoring tool can be designed as longitudinal assessment with only short intervals between measurement points. The reliability of these instruments has been mostly evaluated in observational studies with small samples based on generalizability theory. However, for a standardized use in the pedagogical field, a larger and broader sample is required in order to assess measurement invariance between different participant groups and over time. Therefore, we constructed a DBR, the Questionnaire for Monitoring Behavior in Schools (QMBS) with multiple items to measure the occurrence of specific externalizing and internalizing student classroom behaviors on a Likert scale (1 = never to 7 = always). In a pilot study, two trained raters observed 16 primary education students and rated the student behavior over all items with a satisfactory reliability. In the main study, 108 regular primary school students, 97 regular secondary students, and 14 students in a clinical setting were rated daily over one week (five measurement points). Item response theory (IRT) analyses confirmed the technical adequacy of the instrument and latent growth models demonstrated the instrument's stability over time. Further development of the instrument and study designs to implement DBRs is discussed.

**Keywords:** direct behavior rating; test; sensitivity over time; rating; school; classroom behavior; progress monitoring

## 1. Introduction

Emotional and behavioral problems in students pose a big challenge in classrooms. These problems have been structured traditionally into externalizing and internalizing behavior problems (Achenbach and Edelbrock 1978). Externalizing behavior problems are outwardly directed behaviors, which are a representation of a maladaptive underregulation of cognitive and emotional states (Achenbach and Edelbrock 1978). Meanwhile, internalizing behavior problems typically develop and persist within an individual, and they represent a maladaptive overregulation of cognitive and emotional states (Achenbach and Edelbrock 1978). According to national and international prevalence studies, 10% to 20% of all school-age children and adolescents show these behavioral problems (e.g., Costello et al. 2003). Longitudinal studies focusing on the consequences of students' externalizing and internalizing behavior problems have shown that the aforementioned behavior patterns in the

classroom correlate with academic failure, social exclusion, and delinquency (e.g., Krull et al. 2018; Moffitt et al. 2002; Reinke et al. 2008). In addition, teachers report high levels of stress when they face students' externalizing and internalizing behavior problems in the classroom (e.g., Center and Callaway 1999).

School-based behavioral interventions have been shown to be an efficient way to prevent and decrease the occurrence of externalizing and internalizing behavior problems (e.g., Durlak et al. 2011; Fabiano and Pyle 2018; Waschbusch et al. 2018). However, the effectiveness of these intervention methods increases when intervention planning, implementation, and evaluation are closely linked to school-based assessment practices (Eklund et al. 2009). Two assessment methods have been shown to lead to more effective interventions (Volpe et al. 2010): universal behavior screening and behavior progress monitoring. Universal screening tools identify students who might benefit from a behavior intervention and additionally guide its planning and implementation. Behavioral progress monitoring is used to evaluate an individual student's response to a behavioral intervention. Behavioral progress monitoring data is collected very frequently up to several times a day. It allows teachers to recognize behavioral changes of the students over a short time period, which assists decisions about maintaining or modifying the intervention.

While there are many existing tools that can be used for universal behavior screening (e.g., Daniels et al. 2014; Volpe et al. 2018 for an overview), the development of methods that can be used for behavioral progress monitoring is still in its initial stages. Traditionally, there are two widely used approaches that have been used for school-based behavior assessment: behavior rating scales (BRS) and systematic direct behavior observations (SDO; Christ et al. 2009). BRS usually consists of a pool of items representing specific behaviors that an individual might show. The intensity or frequency of the occurrence of these behaviors are rated on a Likert scale. Therefore, the documentation and interpretation of the behavior occur at the same time. BRS can be completed by multi-informants such as the teachers, the parents, or the individual themself. BRS are an efficient way to measure specific behaviors, since they are easy to understand, complete, and interpret. However, the scores generated by BRS represent a subjective perception of an individual's behavior. SDO, in contrast, represent an objective tool to assess a student's behavior (Volpe et al. 2005). In SDO, the documentation and interpretation of an individual's behavior are usually separated. The observer identifies and defines the behavior of interest, and the observation interval. Afterwards, the targeted behavior will be observed in the relevant interval by using time-sampling methods. Finally, the observation scores are analyzed and interpreted. While this procedure generates objective, reliable, and valid data, it is work and time intensive. Furthermore, observation training is often required. In conclusion, BRS and SDO alone have limitations when collecting behavioral progress monitoring data (Christ et al. 2009).

Direct Behavior Rating (DBR) represents a relatively new assessment method, which allows for progress monitoring measurements over short intervals. DBR is a hybrid form of systematic direct observation and behavior rating scales wherein individuals observe and rate (e.g., on a Likert scale) a behavior in a specific situation immediately afterwards (Chafouleas 2011). In recent years, two DBR forms have been developed and evaluated for progress-monitoring purposes: Single-Item Scales (DBR-SIS) and Multi-Item Scales (DBR-MIS; Volpe and Briesch 2015). DBR-SIS usually targets more global behaviors (e.g., academically engagement, disruptive behavior) and may be the most efficient way to broadly measure a student's overall level of behavioral success. This information could be useful when a student exhibits a broad range of specific problem behaviors that are related to problem behavior in general. However, DBR-SIS has not typically been used to assess specific classroom behaviors (e.g., hand raising), which might be more informative for evaluating a student's response to behavioral intervention. In contrast, DBR-MIS usually includes three to five specific behavior items (e.g., completes classwork in allowed time, starts working independently, turns in assignments appropriately) that operationalize a higher-order behavioral dimension. These more specific items can then be analyzed individually or added up to produce a sum score (Volpe and Briesch 2012).

Previous studies have shown that DBR meets the criteria required for behavioral progress-monitoring. First, DBR is feasible and effective because it does not require extensive materials and the ratings can be completed easily in a few minutes (Chafouleas 2011). Second, DBR is flexible because a broad range of observable behaviors (at both the global and specific levels) can be addressed. Third, DBR is repeatable because the same behavior target can be observed and rated across many observations. Fourth, the psychometric quality of DBR has been supported by a broad range of evaluation studies focusing on the performance of the tool under different measurement conditions (Chafouleas 2011; Christ et al. 2009; Huber and Rietz 2015).

Most DBR studies, both within Germany and the rest of the world, have evaluated its reliability using Generalizability Theory (GT; see Huber and Rietz 2015). Within generalizability theory, which represents a liberalization of Classical Test Theory (CTT), assessments are tied closely to the target populations with respect to the variability of the targeted behaviors. This technique can establish the external validity of a DBR by ensuring that the behavioral targets and evaluation groups are well matched. Most studies were designed along generalizability theory in order to measure the true behavior and investigate potential factors (and their interactions) that might influence the variance in the generated scores (e.g., such as multiple raters and multiple time points). Such studies are necessary to examine the reliability of behavioral assessment and to determine conditions that might increase the reliability (Cronbach et al. 1972). Previous studies found that DBR generates reliable scores by reflecting a large amount of variance explained by the actual student's behavior (e.g., Owens and Evans 2017). However, results from different raters across multiple time points indicate that different persons rate the same behavior differently and that students behave differently across multiple occasions (e.g., Briesch et al. 2010; Volpe and Briesch 2012; Briesch et al. 2014). Therefore, multiple measurement points are necessary to provide a stable score that still is interpretable. Previous generalizability studies showed that valid results are generated within 4 to 20 measurement points, and that fewer measurement points were needed when DBR-MIS was used (e.g., Casale et al. 2015; Volpe et al. 2011; Volpe and Briesch 2012). While GT represents a strong effort to develop reliable testing, validity concerns within the framework of item response theory (IRT) remain.

Even if the results on the psychometric characteristics of DBR are promising, there are still two remaining issues. First, most of the previous studies had small sample groups with five to ten students and three or more raters. Because of theoretical assumptions in generalizability studies, smaller samples are often used, but such a small sample is insufficient for the evaluation of validity and testing technical adequacy of the test itself. For instance, Rasch modelling may require 100 participants, with 250 for high stakes decisions like screenings, diagnoses, or classroom advancement to obtain sufficiently precise parameter estimates (Linacre 1994). It is therefore important to embed evaluation throughout the development process and to use an evaluation sample that is a sufficient size for IRT analyses. Therefore, DBR should be developed in lines of both generalizability theory and IRT, with evaluation embedded throughout development.

Second, measurement invariance of DBR across multiple occasions has not been examined yet. Since DBR was developed for assessment within a problem-solving model, it has to be sensible to behavioral progress (Deno 2005; Good and Jefferson 1998). Only when DBR scores are comparable over time, the results can be used to draw valid conclusions regarding the behavioral progress and responses to behavioral interventions (Gebhardt et al. 2015).

## 2. Present Study

We designed a DBR, the Questionnaire for Monitoring Behavior in Schools (QMBS), based on the principles of an established screening instrument for use in the educational field. While single teacher ratings can be biased, they are still standard practice in educational evaluations. In order to ensure the psychometric quality of the QMBS, we used five measurement points per single rater. Teacher ratings were used to allow for this relatively high number of ratings required per student.

We analyzed our new instrument with an IRT Rasch model and then evaluated the measurement invariance based on gender, migration background, and school level separately. We also used a latent growth model to investigate systematic changes in ratings over time based on these qualities. Our analysis follow five major research questions, described below:

**Question 1:** Are the QMBS results comparable between the first and last measurement point? We expect to find acceptable measurement invariance between these two measurement points.

**Question 2:** Does the QMBS scale fit an IRT Rasch model? We expect the INFIT and OUTFIT values for our items to be within an acceptable range (0.8 and 1.2) for initial test development.

**Question 3:** Does the QMBS scale fit a four factor structure, and is this structure invariant across gender, migration background, and school level at the final measurement point? We expect that the test will perform similarly for all three groupings.

**Question 4:** Do QMBS scores change over time? As we have a brief measurement period, we expect that in a latent growth model, QMBS scores will remain relatively stable, and that participant gender, special education needs, or migration background will not affect any change in scores over time.

**Question 5:** Is there a significant rater effect upon QMBS scores? Due to the objective nature of the test questions, we expect the ICC within raters to remain relatively low.

## 3. Methods

### 3.1. Sample and Data Collection

Forty-one teachers rated 219 students. Teachers selected their own students with external or internal behavior problems in their classes. On average, the teachers rated five students. Therefore, our sample is not a full class sample, but consists only of a subset of students with behavioral problems. Over one week (five measurement points) 108 primary school students (mean age = 9.1 years, SD = 1.2 years), 97 secondary students (mean age = 14.0 years, SD = 1.5 years) and 14 students in a clinical setting (mean age 13.9 years, SD = 1.1 years) were rated. For each student, the same teacher provided the rating at each measurement point. In the school sample 35 students (17%) had been officially diagnosed with special education needs by relevant specialists, 40 students had a migration background (21%), and 145 students were boys (83%). In the clinical sample 5 students had a migration background (36%) and 11 students were boys (79%).

### 3.2. Instrument

We developed a multidimensional DBR-MIS, the Questionnaire for Monitoring Behavior in Schools (QMBS) with different dimensions for externalizing, internalizing, and positive behaviors (Gebhardt et al. 2018). Along the lines of other established screening tools (i.e., the Strengths and Difficulties Questionnaire; Goodman 1997, 2001; Voss and Gebhardt 2017; DeVries et al. 2018), we created a six-dimensional scale divided into three areas, internalizing problems, externalizing problems, and positive behaviors in school. Internalizing problems included the dimensions depressive and anxious behaviors (DAB) and social interactions problems (SIP). Externalizing problems included the dimensions disruptive behavior (DB) and academic engagement problems (AEP). Lastly, positive behaviors in school included the dimensions scholastic behavior (SB) and prosocial behavior (PS).

The QMBS includes 6 dimensions described in Table 1. They are disruptive behavior, academic engagement problems, depressive and anxious behaviors, social interactions problems, scholastic behavior, and prosocial behavior. The dimensions can be further reduced to three categories: internalizing, externalizing, and positive behaviors. Each dimension includes three new items. Each item assesses a single behavior, which can be observed in one school hour. All items had seven categories from 1 (never) to 7 (always), which was suggested by Christ et al. (2009). Additionally, we constructed a new scale about school behavior. All items were unidirectional. All items are presented in the Appendix A in both English and German languages. A short descriptor for each item as well as their categorization is also available in

Table 1. The quesionaire is licensed under the Creative Commons Attribution-NonCommercial-ShareAlike International License 4.0 (Gebhardt et al. 2018).

**Table 1.** Stability of the QMBS categories in a model over all five times are shown with the point-biserial correlation coefficients and the root-mean-square statistics (INFIT and OUTFIT).

| Items | Short Descriptor | Response Level | | | | | | Rout-Mean-Square Statistics | |
|---|---|---|---|---|---|---|---|---|---|
| | | 1 | 2 | 3 | 4 | 5 | 6 | Infit$_{MSQ}$ | Outfit$_{MSQ}$ |
| **Disruptive Behavior** | | | | | | | | | |
| DB01 | Temper | −0.70 | −0.11 | 0.13 | 0.23 | 0.40 | 0.62 | 0.97 | 1.00 |
| DB02 | Disobeys | −0.62 | −0.22 | 0.04 | 0.26 | 0.33 | 0.62 | 1.07 | 1.06 |
| DB03 | Argues | −0.70 | −0.14 | 0.04 | 0.21 | 0.37 | 0.66 | 0.96 | 0.94 |
| **Academic Engagement** | | | | | | | | | |
| AEP04 | Fidgets | −0.67 | −0.09 | −0.01 | 0.16 | 0.29 | 0.58 | 1.05 | 1.09 |
| AEP05 | Quits Early | −0.62 | −0.28 | −0.06 | 0.16 | 0.36 | 0.53 | 1.05 | 1.05 |
| AEP06 | Distracted | −0.54 | −0.39 | −0.25 | −0.03 | 0.20 | 0.68 | 0.90 | 0.85 * |
| **Depressive and Anxious Behaviors** | | | | | | | | | |
| DAB07 | Sad | −0.70 | −0.18 | 0.02 | 0.25 | 0.23 | 0.54 | 0.97 | 0.92 |
| DAB08 | Fearful | −0.67 | 0.10 | 0.08 | 0.21 | 0.29 | 0.42 | 1.07 | 1.09 * |
| DAB09 | Nervous | −0.70 | 0.04 | 0.08 | 0.36 | 0.28 | 0.45 | 0.97 | 0.97 |
| **Social Interaction Problems** | | | | | | | | | |
| SIP10 | Loner | −0.65 | −0.16 | 0.01 | 0.25 | 0.35 | 0.63 | 1.03 | 1.00 |
| SIP11 | Teased | −0.71 | 0.00 | 0.18 | 0.29 | 0.41 | 0.57 | 0.96 | 0.96 |
| SIP12 | Prefers Adults | −0.67 | −0.10 | 0.06 | 0.29 | 0.38 | 0.59 | 1.01 | 1.05 |
| **Scholastic Behavior** | | | | | | | | | |
| SB13 | Participate | −0.41 | −0.14 | 0.01 | 0.06 | 0.35 | 0.39 | 1.49 * | 1.49 * |
| SB14 | Rules | −0.49 | −0.30 | −0.17 | 0.08 | 0.26 | 0.55 | 0.97 | 0.98 |
| SB15 | Concentrates | −0.55 | −0.36 | −0.04 | 0.20 | 0.46 | 0.51 | 0.74 | 0.72 |
| SB16 | Works quietly | −0.48 | −0.35 | −0.17 | 0.10 | 0.34 | 0.59 | 0.80 | 0.80 |
| **Prosocial Behavior** | | | | | | | | | |
| PS17 | Considerate | −0.62 | −0.34 | −0.16 | 0.10 | 0.40 | 0.56 | 1.02 | 1.04 |
| PS18 | Helpful | −0.63 | −0.36 | −0.16 | 0.10 | 0.37 | 0.61 | 0.94 | 0.92 |
| PS19 | Cooperative | −0.63 | −0.33 | −0.13 | 0.17 | 0.38 | 0.58 | 1.04 | 1.04 |

Note: * $p < 0.05$.

Initial Validation and Further Data Collection

In an initial exploratory study, we asked three special school teachers in special schools with children with emotional and social problems to provide an expert-review of the items. Each special schoolteacher rated three children with emotional and social problems. Afterwards, we interviewed the teachers and adapted the items. Next, we tested the new items in a second pilot study with 15 students and two trained raters. The raters had a high compliance rate of spearman's rho = 0.84. Finally, the items were discussed with raters and minor rewording was done.

*3.3. Analyses*

The analyses were carried out with the statistics program R (R Core Team 2013) using the package pairwise (Heine 2014). Here the method of explicit calculation of the item parameters in the fast model was used by the pairwise item comparison (Chopin 1968; Wright and Masters 1982). This method is particularly suitable for determining the sample-invariant item parameters for the calibration of a given item pool for an unidimensional model (Chopin 1968). The pairwise estimator is also suitable for small samples or data sets with missing values (Wright and Masters 1982; Heine and Tarnai 2015; Heine et al. 2018). First, the measurement invariance over the four time points was checked by means of the graphical model test. Second, the item parameters were calculated over all five measurement times and third, the fit of the model was determined at all measurement times using mean square fit

statistics (infit and outfit). The personal parameters were estimated for the respective measurement times using the weighted maximum likelihood method (Warm 1989). For the common item parameters, the point-biserial correlations with the scale value (WLE estimator) are reported as selectivity for the respective measurement time.

## 4. Results

### 4.1. Measurement Invariance between Time Points One and Five

First, graphical model tests confirmed the measurement invariance between time points one and five separately for every dimension (see Figure 1). For this analysis, response categories six and seven were combined, because category seven was rarely used.

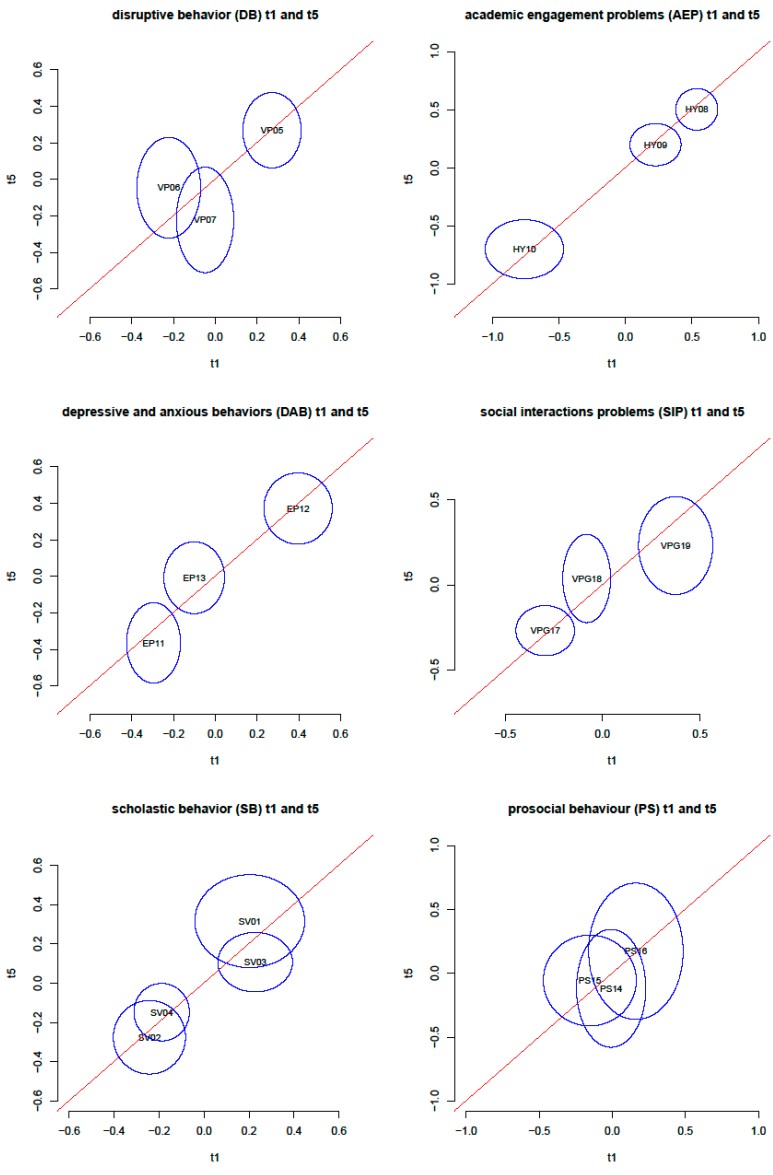

**Figure 1.** Graphical Model-Test with split criterion time points one and five for six scales of disruptive behavior (DB), academic engagement problems (AEP), depressive and anxious behaviors (DAB), social interactions problems (SIP), scholastic behavior (SB), and prosocial behavior (PS) of the QMBS.

### 4.2. Rasch Modeling

In addition, we investigated the model at the item level via infit and outfit. The outfit is the sum of squared standardized residuals and is sensitive to raters and oversights. In contrast, the infit is weighted for information and shows distortions in the sample (such as Guttman Pattern). The results from the root-mean-square statistics (INFIT and OUTFIT) as well as the point-biserial correlation coefficients are presented in Table 1 and indicate that most of the items fit well with the assumptions of the Rasch model (Wright and Masters 1982). Only the item SB13 "Participates in class" shows a low discrimination. Nonetheless, it is still in the range between 0.5 and 1.5, which Linacre (2002) has suggested as an acceptable range for questionnaires. Although, for high-stake tests used to evaluate students, a stricter range of 0.8 to 1.2 is proposed (Wright and Linacre 1994). The WLE reliability, which is comparable to Cronbachs Alpha, was sufficient (DB 0.76, AEP 0.77, DAB 0.70, SIP 0.65, SB 0.77, PS 0.87).

### 4.3. Measurement Invariance at Measurement Point Five

A CFA tested the hypothesized 6-factor structure using data from measurement point five. Initial results produced an unacceptable initial fit with RMSEA = 0.10, CFI = 0.87, and SRMR = 0.09. However, a modified model produced an acceptable fit, with RMSEA = 0.08, CFI = 0.92, and SRMR = 0.06. These minor modifications included dropping item SB13 because of a low loading (0.34), and allowing item SP11 to cross load onto DB.

Measurement invariance was also assessed at measurement point 5 across gender, migration background, and school level. For the school level assessment only, the clinical population was excluded. All analyses of invariance were conducted in Mplus 7.4. We assessed weak invariance by comparing the fits of the configural to metric models and strong invariance by comparing fits of the metric and scalar models (see the procedure described by Dimitrov 2017). We used the threshold of $\Delta$CFI < 0.010 to indicate a significant change in fit.

Results upheld both weak invariance in all cases with $\Delta$CFI < 0.010. Similarly, strong invariance was found for gender and migration background, $\Delta$CFI < 0.010. However, strong invariance was not upheld for school level, $\Delta$CFI = 0.012. We proceeded to test for partial invariance by individual freeing intercepts with the greatest effect on $\chi^2$ until the difference between the metric and scalar model was under threshold of $\Delta$CFI < 0.010. We used the standard for partial invariance of fewer than 20% of freed intercepts and loadings (see Levine et al. 2003). The freeing of a single intercept (AEP 05) resulted in a net $\Delta$CFI = 0.005, under the threshold of 0.010. As this represented only 3% of the model's intercepts and loadings, we concluded that the instrument possessed sufficient partial invariance across grade level, and that overall comparisons across gender, migration background, and grade level are meaningful with the instrument.

### 4.4. Latent Growth Models

Four separate latent growth models were calculated to estimate the change in individual WLE scores over time. In these models disruptive behavior (DB) and academic engagement problems (AEP) were collapsed into a single externalizing factor, and depressive and anxious behaviors (DAB), social interactions problems (SIP) were collapsed into a single internalizing factor. In the models, gender, migration background, attending the secondary school, and attending the clinical school were used as predictors for the slope and intercept parameters in these models. As Table 2 describes, model fits were good in all cases. Table 3 describes the intercept and slope values for the models, as well as the path loadings. Overall slopes were insignificant, but overall intercepts differed from zero. The distribution of slopes can be seen in Figure 2, where the slopes are all close to zero. This indicates that individual persons did not change much on average over time. Furthermore, boys had higher levels of externalizing than girls, and children with a migration background also had a higher level of externalizing behavior. The clinical subsample showed a higher slope of internalizing as well

as a lower slope in scholastic behavior, indicating more problematic development within a short time frame. However, the clinical subsample displayed a higher intercept in scholastic behavior and internalizing. Girls had higher scores on prosocial behavior and scholastic behavior. The clinical population displayed a significant effect on the slopes for internalizing, scholastic behavior, and prosocial behavior. This may indicate a systematic change in these values for this population.

**Table 2.** Fit values for the Latent Growth Models.

| Dimension | RMSEA (90% CI) | CFI | SRMR |
|---|---|---|---|
| Externalizing (AEP + DB) | 0.00 (0.00–0.01) | 1.00 | 0.02 |
| Internalizing (DAB + SIP) | 0.00 (0.00–0.03) | 1.00 | 0.01 |
| Prosocial Behavior | 0.06 (0.02–0.09) | 0.99 | 0.03 |
| Scholastic Behavior | 0.00 (0.00–0.04) | 1.00 | 0.02 |

**Table 3.** Relevant values from latent growth models.

| Dimension | Model | Intercept | Slope |
|---|---|---|---|
| Externalizing (AEP + DB) | Overall | −2.02 *** | −0.03 |
| | Gender | 0.95 * | −0.02 |
| | Migration Status | 0.48 * | 0.03 |
| | Secondary School | 0.08 | −0.01 |
| | Clinical School | 0.18 | 0.06 |
| Internalizing (DAB + SIP) | Overall | −0.61 * | −0.03 |
| | Gender | −0.01 | −0.04 |
| | Migration Status | 0.00 | 0.05 |
| | Secondary School | 0.24 | 0.06 * |
| | Clinical School | 0.43 *** | 0.08 **** |
| Prosocial Behavior | Overall | 1.09 *** | 0.02 |
| | Gender | −1.42 *** | 0.08 |
| | Migration Status | −0.55 | −0.06 |
| | Secondary School | −0.2 | −0.07 |
| | Clinical School | −0.03 | −0.13 * |
| Scholastic Behavior | Overall | 2.63 *** | −0.03 |
| | Gender | −0.74 *** | 0.01 |
| | Migration Status | −0.35 | 0.01 |
| | Secondary School | −0.04 | −0.01 |
| | Clinical School | 0.51 * | −0.13 ** |

Note: Secondary School and Clinical School are dummy coded categories for school setting. * significant at $p < 0.05$; ** significant at $p < 0.01$; *** significant at $p < 0.001$.

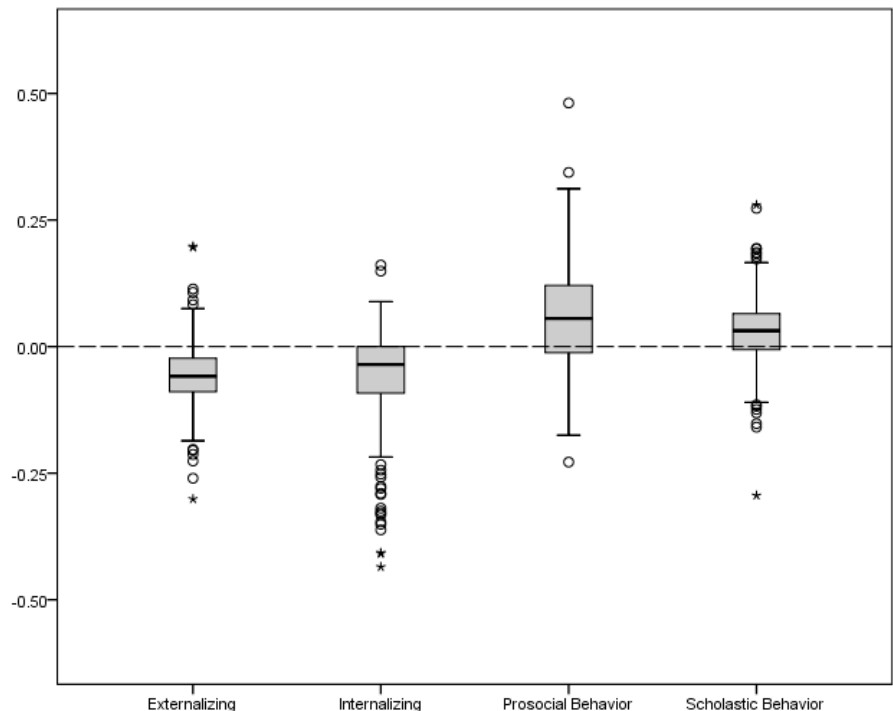

**Figure 2.** Individual unstandardized slopes for the latent growth models.

*4.5. Intraclass Correlations of Different Raters*

The intraclass correlations based on rater remained low for the same four subscales at measurement point five, $ICC_{externalizing} < 0.01$, $ICC_{internalizing} = 0.07$, $ICC_{prosocial\ behavior} = 0.01$, and $ICC_{scholastic\ behavior} < 0.01$.

## 5. Discussion

This study demonstrated an approach complementary to generalizability theory to examine the item characteristics and the stability of QMBS ratings across occasions. Our relatively large sample allowed more detailed IRT analysis than the smaller samples of generalizability theory-based DBR development and assessment. This represents a significant addition to previous DBR assessment and development techniques. With such advances in DBR techniques, we provide a framework for future studies to improve the assessment of both the reliability and validity of DBR techniques. This provides a significant step forward to DBR research within both Germany and around the world.

Our QMBS had a high compliance rate by two trained raters in a pilot study and showed in IRT analyses invariance over five measurement points and satisfactory reliability on the item level. Results from the CFA confirmed the overall factor structure parallels the structure of past research (e.g., SDQ; Goodman et al. 2010), with only minor modifications. Invariance tests revealed its applicability to diverse groups based on gender, migration background, and school level. Results of the latent growth models confirmed the overall stability of the scores across all five measurement points. The intraclass correlations based on rater indicate that there was little effect of rater bias on the overall WLE scores.

The assessment of invariance for the QMBS is an important early step in the development of the scale. This invariance means that the test performs comparably for our primary, secondary, and clinical student samples, as well as both genders, for learners with SEN, and for those with a migration background. Similarly, it performed comparably across the multiple measurement points. This means that over time, the results from each item may be compared to results at a previous time point. More generally, the CFA and invariance results mean that meaningful comparisons can be made using the sum scores for the different dimensions (Dimitrov 2017). This is an important requirement for any scale which uses repeated measurements which track change in behavior over time.

## 5.1. Limitations and Future Work

We did not provide any experimental treatments, and every teacher rated their own students individually. Therefore, it was expected that there would be no difference between school levels or types or across measurement points. Most teachers provided ratings that were in the center category and their ratings remained stable. The highest category, seven (always), was so rarely used, that we needed to combine this category with a lower rating for the IRT model. Therefore, our instrument cannot compare different groups of students to one measurement point, but it is instead more suited to measure the individual change over time. Additional studies are needed to measure the sensitivity for the change in behavior over time. For this, studies with interventions are needed to explain the behavior over short and long time periods. Lastly, normative values should be found across a large, random sample which includes learners across a diverse group of schools. Another caveat is that our design did not allow for a detailed analysis of rater effects (e.g., rater severity, rater drift over time), which can affect longitudinal ratings substantially. Especially in order to disentangle rater effects from item stability/sensitivity to interventions over time, more complex designs are needed.

## 5.2. Implications for Research and Practice

Our study demonstrated the value of an IRT approach to DBR. Specifically, that such an approach can help validate the test items. Furthermore, items can be assessed for invariance across a number of groupings. This approach is sorely needed as a compliment to DBR approaches focusing solely on generalizability theory.

## 6. Conclusions

It is possible to develop direct behavior ratings methods in a rigorous manner. We conclude that the behavioral measurements with constant items over short measurement periods with the same teacher are stable and individually evaluable. More work to measure the sensitivity to change and responses to treatment is needed to further develop these methods.

**Ethical Statement:** All subjects gave their informed consent for inclusion before they participated in the study. The study was conducted in accordance with the dean of the Faculty of Rehabilitation Science, Technical University of Dortmund. An additional ethics approval was not required for this study as per Institution's guidelines and national regulations.

**Author Contributions:** Conceptualization, M.G. and G.C.; methodology, M.G.; validation, M.G., J.M.D. and J.-T.K.; formal analysis, M.G. and J.M.D.; investigation, M.G. and J.J.; resources, M.G., J.M.D. and J.J.; data curation, M.G. and J.M.D.; writing—original draft preparation, M.G., J.M.D. and G.C.; writing—review and editing, M.G., J.J., A.G., J.-T.K.; visualization, J.J.; supervision, M.G. and J.-T.K.; project administration, M.G.

**Funding:** This research received no external funding.

**Conflicts of Interest:** The authors declare no conflict of interest.

## Appendix A

*Appendix A.1 Instructions and procedures for the "Questionnaire for Monitoring Behavior in Schools" (QMBS)*

This questionnaire covers the behavior of pupils during a clearly defined timeframe (e.g., one day of instruction). The goal is to get subsequent ratings in comparable time frames (e.g., a math lesson or the whole day). With such a series, it is possible to map the pattern of behavior of the pupil.

You are free to decide upon the observation timeframe, except that they are comparable in terms of length and didactic content. Also, the same person should rate the same pupil at each timeframe.

If you encounter items that you can not rate based on observations within the timeframe, check the value of the previous day or leave this item blank.

*Appendix A.2 Questionnaire for Monitoring Behavior in Schools (QMBS)*

| Nr. | Items | Never | | | | | | Always |
|---|---|---|---|---|---|---|---|---|
| | Externalizing Behavior | | | | | | | |
| | Disruptive Behavior (DB) | | | | | | | |
| 1 | Has temper tantrums or a hot temper, has a low frustration tolerance. | 1 | 2 | 3 | 4 | 5 | 6 | 7 |
| 2 | Disobeys rules and does not listen to the teacher | 1 | 2 | 3 | 4 | 5 | 6 | 7 |
| 3 | Argues with classmates/provokes classmates with his/her behavior | 1 | 2 | 3 | 4 | 5 | 6 | 7 |
| | Academic Engagement Problems (AEP) | | | | | | | |
| 4 | Fidgets or squirms, is restless/overactive | 1 | 2 | 3 | 4 | 5 | 6 | 7 |
| 5 | Frequently quits tasks early | 1 | 2 | 3 | 4 | 5 | 6 | 7 |
| 6 | Easily distracted | 1 | 2 | 3 | 4 | 5 | 6 | 7 |
| | Internalizing Behavior | | | | | | | |
| | Depressive and anxious behaviors (DAB) | | | | | | | |
| 7 | Seems worried, sad, or depressed | 1 | 2 | 3 | 4 | 5 | 6 | 7 |
| 8 | Seems Fearful | 1 | 2 | 3 | 4 | 5 | 6 | 7 |
| 9 | Seems nervous. | 1 | 2 | 3 | 4 | 5 | 6 | 7 |
| | Social interactions problems (SIP) | | | | | | | |
| 10 | Works/plays mostly alone, prefers to be alone | 1 | 2 | 3 | 4 | 5 | 6 | 7 |
| 11 | Teased or bullied by classmates, easily provoked | 1 | 2 | 3 | 4 | 5 | 6 | 7 |
| 12 | Gets along better with adults than with other children | 1 | 2 | 3 | 4 | 5 | 6 | 7 |
| | Positive Behavior in School | | | | | | | |
| | Scholastic Behavior (SB) | | | | | | | |
| 13 | Participates in class | 1 | 2 | 3 | 4 | 5 | 6 | 7 |
| 14 | Follows rules for speaking in class (i.e., raises hand) | 1 | 2 | 3 | 4 | 5 | 6 | 7 |
| 15 | Concentrates on his/her schoolwork | 1 | 2 | 3 | 4 | 5 | 6 | 7 |
| 16 | Works quietly at his/her desk/does not refuse assignments | 1 | 2 | 3 | 4 | 5 | 6 | 7 |
| | Prosocial Behavior (PS) | | | | | | | |
| 17 | Considerate of other people's feelings | 1 | 2 | 3 | 4 | 5 | 6 | 7 |
| 18 | Helpful to others | 1 | 2 | 3 | 4 | 5 | 6 | 7 |
| 19 | Cooperative in partner and group situations | 1 | 2 | 3 | 4 | 5 | 6 | 7 |

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
