# Peer review of "Measurement Invariance of a Direct Behavior Rating Multi Item Scale across Occasions"

_socsci, doi:10.3390/socsci8020046_

Round 1

Reviewer 1 Report

More details (i.e., demographic, SES background, ethnicity, diagnoses, related services) would be needed for the participants. The readers would be interested to learn how and for what reason the students and their teachers were selected for the study. 

Gender, migration status and school setting were chosen to be the three categories of measurement invariance. Data analysis should be reported in accordance with these three main variables (i.e., school setting instead of the secondary and clinical school in Table A3). In additon, more discussion is needed about the findings regarding these three invariances. 

Table numbers need to be consistent in the text as shown in table captions (i.e.,A3 instead of 3). 

Author Response

Dear Reviewer 1

General Comments

We would like to thank both reviewers for their hard work in reviewing our article. Their work has provided significant improvements to the article. Below we detail our responses to the points from both reviewers.

Reviewer 1

More details (i.e., demographic, SES background, ethnicity, diagnoses, related services) would be needed for the participants. The readers would be interested to learn how and for what reason the students and their teachers were selected for the study. 

We provide additional data on lines 176 to 177 about the ages of the participants, “108 primary school students (mean age = 9.1 years, SD = 1.2 years), 97 secondary students (mean age = 14.0 years, SD = 1.5 years) and 14 students in a clinical setting (mean age 13.9 years, SD = 1.1 years)“ Beyond the number of participants with migration background (21%), we are unable to provide additional SES and ethnicity data, as such data was not collected. Similarly, teacher s did not provide specifics regarding diagnoses for SEN. However, such diagnoses would be officially made by the relevant administrators and staff within the regular school, and thus reflect official guidelines. We have appended line 180 to reflect that the diagnoses were official “35 students (17%) had been officially diagnosed with special education needs by relevant specialists“

Gender, migration status and school setting were chosen to be the three categories of measurement invariance. Data analysis should be reported in accordance with these three main variables (i.e., school setting instead of the secondary and clinical school in Table A3).

We clarify in the figure caption that Secondary School and Clinical School are the dummy coded labels for the three possible values of school setting, “Secondary School and Clinical School are dummy coded categories for school setting.

In additon, more discussion is needed about the findings regarding these three invariances. 

We have added a paragraph starting at line 340 which stresses the importance and value of the invariance checks. We have made several other minor edits to the discission section to further highlight this. Starting line 340: “The assessment of invariance for the QMBS is an important early step in the development of the scale. This invariance means that the test performs comparably for our primary, secondary, and clinical student samples, as well as both genders, for learners with SEN, and for those with a migration background. Similarly, it performed comparably across the multiple measurement points. This means that over time, the results from each item may be compared to results at a previous time point. More generally, the CFA and invariance results means that that meaningful comparisons can be made using the sum scores for the different dimensions (Dimitrov, 2017). This is an important requirement for any scale which uses repeated measurements which track change in behavior over time.”

Table numbers need to be consistent in the text as shown in table captions (i.e.,A3 instead of 3). 

We have renamed our tables and figures appropriately to match the text. We no longer label any table or figure with an “A” prefix.

Thank you for you notes. We feel they have substantially improved the quality of our article.

Best regards

Reviewer 2 Report

Thank you for the opportunity to review the manuscript: Measurement Invariance of a Direct Behavior Rating 2 Multi Item Scale across Occasions. The authors are focused on development and extension of a tool to progress monitor social behavior. Behavioral progress monitoring is an important topic in the area of education, especially the progress monitoring of students with or at-risk of emotional and behavioral disorders and related disabilities. The authors have extended work in the United States on behavioral progress monitoring (e.g., Volpe et al. and others). This work is important and we have adapted a similar line of research in our own work in Texas around positive behavioral support and monitoring responsiveness to intervention. My recommendation to the editor is acceptance pending the following recommendations:

1.     The authors provide a framework for the study primarily based on research in the US. I would recommend adding a paragraph starting on line 113 summarizing the specific state of the research on DBRs conducted in Germany since it is where the study occurred and is the sample.

2.     Add a paragraph on validity or add some framing sentences about validity (e.g., Messick) - perhaps when you discuss generalizability theory.  This is really a concurrent/predictive validity study between DBR and SDQ or discuss the implications for validity in the discussion section.

3.     I would do the same in the discussion- implications for furthering the larger research base on DBR and then implications for research on DBRs in Germany as it relates to future research and practice.

4.     Move the text from lines 115 to 129 and integrate it into Section 2. Present Study on line 131.

5.     I would recommend listing out the research questions by number: 1. What is the concurrent validity of the DBR with the SDQ? Etc. and then organizing the first part of the discussion by each finding/research question. I usually recommend that each analysis should have it’s own research question so there is congruence between the question and the analysis used to answer the question which makes it much easier for the reader to follow when there are multiple analyses.

Methods

 (My recommendations are based on APA- it might be this journal uses a different format so double check with the journal editor)

1.     Move the description of the sample (3.1) to come first in the methods before Materials and Methods.

2.     The description of the data analysis should go under a section header: Data Analysis at the end of the methods. In this section describe the data plan that was used.

3.     I wouldn’t put Materials and Methods- Just methods should be ok as a header for APA.

4.     If the scales are short- I would recommend adding a table with the items and scale or add the actual items to table A1. Or is this what you are doing in the Appendix? Questionnaire for Monitoring Behavior in Schools (QMBS)? If this is the DBR scale- move it as a table into the methods under instrumentation or measures. The table needs to be APA formatted also.

5.     If the QMBS is the DBR- make sure this is clear in the write up- consider instead of DBR something like DBR-QMBS throughout the paper so the reader is clear this is the multi-item DCR being talked about.

6.     On Table A1- put additional sub-headers labeling the construct or subscales under the item column.

7.     Do the same for the SDQ- I am not clear- was the constructs formed ones you created or subscales used in the SDQ? The SDQ focuses on: emotional symptoms, conduct problems, hyperactivity/inattention, peer relationships, and prosocial behavior. It isn’t coming through why you constructed your own constructs given the instrument already has valid subscales.

8.     Add a subheader- Data Collection Procedures, and describe how the data was collected, number of data points, reliability checks, interobserver agreement, etc.

9.     It isn’t clear how many data points were collected for DBRs? In the discussion it indicated 5. How were they were aggregated across the items and scales for analysis?

10.  Also- how long or how many weeks did it take to collect the 5 data points? When was the SDQ administered?

11.  Under Instrument- describe the DBR and SDQ with subheaders and provide the reliability and validity for each.

Results

1.     Consider adding a table of descriptive statistics for each variable.

Discussion

1.     The “So what?” or "take home" messages are not coming through in the discussion. Add a paragraph to remind the reader why the focus on measurement invariance is important and what is demonstrated by it from a validity perspective, then move into the findings by research question. Put subheaders by research question if needed.

2.     Add a “Limitations” Section

3.     Add a “Implications for Research and Practice” section.

Thank you for the opportunity to review the study. The research represented is important to the field in examining the validity of progress monitoring tools for individuals with social and emotional problems.

Author Response

General Comments

We would like to thank both reviewers for their hard work in reviewing our article. Their work has provided significant improvements to the article. Below we detail our responses to the points from both reviewers.

Reviewer 2

1.     The authors provide a framework for the study primarily based on research in the US. I would recommend adding a paragraph starting on line 113 summarizing the specific state of the research on DBRs conducted in Germany since it is where the study occurred and is the sample.

We have added text to line 112 to clarify that Huber & Rietz (2015) describe the state of DBR research in Germany and the rest of the world to be similarly ground within Generalizability Theory, “Most DBR studies, both within Germany and the rest of the world, have evaluated its reliability using Generalizability Theory (GT; see Huber & Rietz, 2015)“

2.     Add a paragraph on validity or add some framing sentences about validity (e.g., Messick) - perhaps when you discuss generalizability theory.  This is really a concurrent/predictive validity study between DBR and SDQ or discuss the implications for validity in the discussion section.

We have added a connecting sentence on line 130 to clarify that one major reason for applying IRT techniques to DBR is validity, which we discuss in the coming two paragraphs, “While GT represents a strong effort to develop reliable testing, validity concerns  within the framework of item response theory (IRT) remain.“

3.     I would do the same in the discussion- implications for furthering the larger research base on DBR and then implications for research on DBRs in Germany as it relates to future research and practice.

We add a the critical sentence at line 329 to strengthen our statement about improved validity, and its application in Germany and abroad, “With such advances in DBR techniques, we provide a framework for future studies to improve the assessment of both the reliability and validity of DBR techniques. This provides a significant step forward to DBR research within both Germany and around the world.“

4.     Move the text from lines 115 to 129 and integrate it into Section 2. Present Study on line 131.

We feel moving these 15 lines of text breaks up the logical flow of the literature review. As these lines focus mostly on previous research, we did not move these descriptions. However, see below (#5) regarding the revision of the present study section.

5.     I would recommend listing out the research questions by number: 1. What is the concurrent validity of the DBR with the SDQ? Etc. and then organizing the first part of the discussion by each finding/research question. I usually recommend that each analysis should have it’s own research question so there is congruence between the question and the analysis used to answer the question which makes it much easier for the reader to follow when there are multiple analyses.

We provide research questions in the present study section starting on line 158 to guide the reader through our results. Included in each research question is a pertinent specific prediction:

“ Question 1: Are our DBR scale results comparable between the first and last measurement point? We expect to find acceptable measurement invariance between these two measurement points.

Question 2: Does our DBR scale fit an IRT Rasch model. We expect the INFIT and OUTFIT values for our items to be within an acceptable range (0.8 and 1.2) for initial test development.

Question 3: Does our DBR scale fit a four factor structure, and is this structure invariant across gender, migration background, and school level at the final measurement point. We expect that the test will perform similarly for all three groupings.

Question 4: Do the DBR scores change over time? As we have a brief measurement period, we expect that in a latent growth model, DBR scores will remain relatively stable, and that participant gender, special education needs, or migration background will not affect any change in scores over time.

Question 5: Is there a significant rater effect upon scores? Due to the objective nature of the test questions, we expect that the ICC within raters to remain relatively low.”

Methods

 (My recommendations are based on APA- it might be this journal uses a different format so double check with the journal editor)

1.     Move the description of the sample (3.1) to come first in the methods before Materials and Methods.

Changed.

2.     The description of the data analysis should go under a section header: Data Analysis at the end of the methods. In this section describe the data plan that was used.

Changed.

3.     I wouldn’t put Materials and Methods- Just methods should be ok as a header for APA.

Changed

4.     If the scales are short- I would recommend adding a table with the items and scale or add the actual items to table A1. Or is this what you are doing in the Appendix? Questionnaire for Monitoring Behavior in Schools (QMBS)? If this is the DBR scale- move it as a table into the methods under instrumentation or measures. The table needs to be APA formatted also.

The items do not fit into Table 1 in their entirety; however, we have updated Table 1 to provide a short description of each item as well as the names of each category. The full questionnaire is still available in the Appendix, where the full item text can be found. We have not changed the format the QMBS in the appendix, as this reflects the actual format of the questionnaire used.

5.     If the QMBS is the DBR- make sure this is clear in the write up- consider instead of DBR something like DBR-QMBS throughout the paper so the reader is clear this is the multi-item DCR being talked about.

We have corrected this issue, and refer to our instrument specifically as the QMBS throughout, and we now use DBR only in the general sense of all direct behavior ratings.

6.     On Table A1- put additional sub-headers labeling the construct or subscales under the item column.

Done. See additional comments from point Methods 4.

7.     Do the same for the SDQ- I am not clear- was the constructs formed ones you created or subscales used in the SDQ? The SDQ focuses on: emotional symptoms, conduct problems, hyperactivity/inattention, peer relationships, and prosocial behavior. It isn’t coming through why you constructed your own constructs given the instrument already has valid subscales.

Our scale does not the SDQ, nor is it based specifically on SDQ items. However, some of our dimensions do mirror the SDQ. We have removed the reference to the SDQ in section in Lines 195 to 200  to clarify that the scale is separate from the SDQ, “The QMBS includes 6 dimensions described in Table 1. They are disruptive behavior, academic engagement problems, depressive and anxious behaviors, social interactions problems, scholastic behavior and prosocial behavior. The dimensions can be further reduced to 3 categories: internalizing, externalizing, and positive behaviors. Each dimension includes three new items. Each item assesses a single behavior, which can be observed in one school hour. All items had seven categories from 1 (never) to 7 (always), which was suggested by Christ, Riley-Tillman und Chafouleas (2009). Additionally, we constructed a new scale about school behavior. All items were unidirectional.“

8.     Add a subheader- Data Collection Procedures, and describe how the data was collected, number of data points, reliability checks, interobserver agreement, etc.

We have added the subheader for section 3.2.1 which includes the initial data collection, validation, and interrater agreement.

9.     It isn’t clear how many data points were collected for DBRs? In the discussion it indicated 5. How were they were aggregated across the items and scales for analysis?

Participants were rated 5 times, see line 178. We have added to the sample subheader “Data collection” so that this information is more easily located.

10.  Also- how long or how many weeks did it take to collect the 5 data points? When was the SDQ administered?

They were rated each day over the course of one school week, see line 178

11.  Under Instrument- describe the DBR and SDQ with subheaders and provide the reliability and validity for each.

As stated above we removed our reference to the SDQ in the instrument section. Section 3.2.1 now describes the initial reliability and validity assessments for the QMBS that were performed during test development.

Results

1.     Consider adding a table of descriptive statistics for each variable.

As our sample is not a normative one, we are hesitant to provide a list of descriptive statistics for each item. We now note in our future research the importance of developing such a data set for the future development of the test. (See response discussion point #3), line 358, “ Lastly, normative values should be found across a large, random sample which includes learners across a diverse group of schools.“

Discussion

1.     The “So what?” or "take home" messages are not coming through in the discussion. Add a paragraph to remind the reader why the focus on measurement invariance is important and what is demonstrated by it from a validity perspective, then move into the findings by research question. Put subheaders by research question if needed.

We added a paragraph summarizing the invariance results starting at line 340, “The assessment of invariance for the QMBS is an important early step in the development of the scale. This invariance means that the test performs comparably for our primary, secondary, and clinical student samples, as well as both genders, for learners with SEN, and for those with a migration background. Similarly, it performed comparably across the multiple measurement points. This means that over time, the results from each item may be compared to results at a previous time point. More generally, the CFA and invariance results means that that meaningful comparisons can be made using the sum scores for the different dimensions (Dimitrov, 2017). This is an important requirement for any scale which uses repeated measurements which track change in behavior over time.“

2.     Add a “Limitations” Section

We have added the subheaded for limitations at 5.1, on line 349

3.     Add a “Implications for Research and Practice” section.

We have added this section starting at line 364, “5.2. Implications for Research and Practice. Our study demonstrated the value of an IRT approach to DBR. Specifically, that such an approach can help validate the test items. Furthermore, items can be assessed for invariance across a number of groupings. This approach is sorely needed as a compliment to DBR approaches focusing solely on generalizability theory.”

Thank you for the opportunity to review the study. The research represented is important to the field in examining the validity of progress monitoring tools for individuals with social and emotional problems.

Thank you for you detailed notes. We feel they have substantially improved the quality of our article.
